# Enhanced Photocatalytic Degradation of Organic Dyes via Defect-Rich TiO_2_ Prepared by Dielectric Barrier Discharge Plasma

**DOI:** 10.3390/nano9050720

**Published:** 2019-05-09

**Authors:** Yanqin Li, Wei Wang, Fu Wang, Lanbo Di, Shengchao Yang, Shengjie Zhu, Yongbin Yao, Cunhua Ma, Bin Dai, Feng Yu

**Affiliations:** 1Key Laboratory for Green Processing of Chemical Engineering of Xinjiang Bingtuan, School of Chemistry and Chemical Engineering, Shihezi University, Shihezi 832003, China; liyanqin112@163.com (Y.L.); wangw@shzu.edu.cn (W.W.); shengchao.yang@shzu.edu.cn (S.Y.); zsj97262724@163.com (S.Z.); yongbinyao0321@sina.com (Y.Y.); mchua@shzu.edu.cn (C.M.); db_tea@shzu.edu.cn (B.D.); 2School of Environmental Science and Engineering, Shanghai Jiao Tong University, Shanghai 200240, China; wangfu@sjtu.edu.cn; 3College of Physical Science and Technology, Dalian University, Dalian 116622, China; dilanbo@163.com

**Keywords:** titanium dioxide, plasma, photocatalytic degradation, organic dye, defects

## Abstract

The dye wastewater produced in the printing and dyeing industry causes serious harm to the natural environment. TiO_2_ usually shows photocatalytic degradation of dye under the irradiation ultravilet light rather than visible light. In this work, a large number of oxygen vacancies and Ti^3+^ defects were generated on the surface of the TiO_2_ nanoparticles via Ar plasma. Compared with pristine TiO_2_ nanoparticles, the as-obtained Ar plasma-treated TiO_2_ (Ar-TiO_2_) nanoparticles make the energy band gap reduce from 3.21 eV to 3.17 eV and exhibit enhanced photocatalytic degradation of organic dyes. The Ar-TiO_2_ obtained exhibited excellent degradation properties of methyl orange (MO); the degradation rate under sunlight irradiation was 99.6% in 30 min, and the photocatalytic performance was about twice that of the original TiO_2_ nanoparticles (49%). The degradation rate under visible light (λ > 400 nm) irradiation was 89% in 150 min, and the photocatalytic performance of the Ar-TiO_2_ was approaching ~4 times higher than that of the original TiO_2_ nanoparticles (23%). Ar-TiO_2_ also showed good degradation performance in degrading rhodamine B (Rho B) and methylene blue (MB). We believe that this plasma strategy provides a new method for improving the photocatalytic activity of other metal oxides.

## 1. Introduction

Printing and dyeing wastewater has caused serious pollution to the environment [1,2], and it is necessary to seek an effective treatment of printing and dyeing wastewater [3]. Semiconductor photocatalytic technology has attracted much attention due to its excellent application prospects in the fields of sewage treatment, air purification, cleaning and sterilization, and solar energy conversion [4,5,6]. It is also an advanced technology that uses solar energy to carry out chemical reactions in a mild environment [7,8,9]. Compared with the traditional pollutant treatment methods, it has the characteristics of mild reaction conditions, complete mineralization of pollutants, high efficiency and high stability, and no secondary pollution. Therefore, it is particularly important to construct an efficient photocatalyst with visible light response and to conduct an in-depth study of its mechanism.

Recently, photocatalysts have been catalytically active substances that can be used in the environment, including water or air purification, water splitting to produce hydrogen, carbon dioxide catalytic reduction and antibacterial action [10,11,12,13,14,15]. Among semiconductor materials, titanium dioxide (TiO_2_) is widely used as a photocatalyst due to its good oxidation and hydrophilic properties, excellent chemical durability and low cost [16,17,18,19]. However, TiO_2_ can only be activated by ultraviolet (UV) light (λ < 380 nm) due to its wide band gap (3.0–3.2 eV). UV light contains about 5% of solar energy reaching the land surface [20,21], and it is crucial to increase the photoabsorption range of TiO_2_ to improve solar energy utilization. TiO_2_ materials can achieve excellent optical absorption properties by doping [22,23,24,25,26,27].

To improve the photocatalytic performance of TiO_2_, some transition metal oxides were introduced into TiO_2_, such as FeO_x_ [28], CuO_x_ [29,30], NiO [31], CeO_2_ [32] and ZnO [33]. Metal doping could make electrons easier to excite and reduce the recombination of electron-hole pairs of TiO_2_ [34,35,36]. Furthermore, enhanced photo-response could be obtained by TiO_2_ with doping non-metals, such as N [37], C [38], F [39] and S [40]. In addition, the defects generated on the surface of TiO_2_ are able to effectively hinder the recombination of photogenerated electron-hole pairs. Surface defects of TiO_2_, for instance, Ti^3+^ defects and oxygen vacancies, promote the formation of the original intermediate band, which plays an important role in photoelectron capture [41].

Low temperature plasma can cause defects and oxygen vacancies on the catalyst surface to improve the catalytic efficiency [42,43,44]. The effect of treating the catalyst with low temperature plasma under different atmospheres is different. Nanowires and nanoparticles of TiO_2_ treated under a hot hydrogen atmosphere have better photocatalytic degradation of dye properties because of the presence of oxygen vacancies and Ti^3+^ forming an intermediate layer [45,46]. The application of oxygen and nitrogen plasma reduced the TiO_2_ powder and produced an intermediate state, resulting in an increase in its optical activity in the visible region [47]. The trapping of charge carriers, decreases the electron-hole recombination rate. However catalyst mechanisms, irradiation conditions and defects have not been fully explained.

Herein, we successfully prepared TiO_2_ nanoparticles by titanium alkoxide hydrolysis method, and etched them in Ar atmosphere with low temperature dielectric barrier discharge (DBD) plasma [48,49]. The results showed that the plasma-etched TiO_2_ nanoparticles had more oxygen vacancies and Ti^3+^ defects than of the original TiO_2_, and thus have excellent photocatalytic degradation properties under sunlight. We believe that DBD plasma provides a new strategy for etching the surface of catalysts.

## 2. Experimental Section

### 2.1. Sample Preparation

The TiO_2_ photocatalyst was prepared using an alkoxide hydrolysis process with titanium n-butoxide (Ti (OC_4_H_9_)_4_, *AR*, Macklin, Shanghai, China) used as the precursor for TiO_2_. 3 g of titanium n-butoxide was added to 100 g of ethanol (C*_2_*H*_5_*OH, AR, Fuyu Fine Chemical Co., Ltd., Tianjin, China) to obtain a mixed solution A; 5 g of deionized water was dissolved in 100 mL of absolute ethanol and stirred for 30 min to make it evenly mixed to obtain a homogeneous solution B; then the solution A was slowly dropped into solution B, and a white precipitate after about 5 min. The solution obtained was stirred vigorously for 1 h, aged for 12 h, and then the white precipitate was filtered, washed 4 times with the alcohol-water mixture, dried in a drying oven at 80 °C for 24 h, and then ground to obtain TiO_2_ nanoparticles. The dried powders were then calcined in air for 5 h with the temperature ranging from 30 to 450 °C. The pristine TiO_2_ obtained was separately subjected to plasma treatment in a dielectric barrier discharge (DBD) plasma reactor at a voltage of 50 V, a current of 1.5 A, and an Ar atmosphere for 20 min to obtain TiO_2_ (Ar-TiO_2_).

### 2.2. Catalyst Characterization

The catalyst crystal structure was identified by X-ray diffractometer (XRD) with Cu-Kα (40 kV, 40 mA, k = 1.5406 Å and 2θ range from 10–90°) radiation (Karlsruhe, Baden-Württemberg, German). The Raman spectrum was measured by Raman spectroscopy using a Renishaw inVia (Renishaw, London, UK) with a laser power of 5 mW and a laser excitation of 532 nm (wavenumber range 100–3200 cm^−1^). Using KBr as diluents, the Fourier transforms infrared spectra (FTIR) of the samples were collected with a Thermo spectrum system (Thermo Fisher Scientific, Massachusetts, MA, USA). The samples were subjected to X-ray photoelectron spectroscopy (XPS, Thermo Scientific Escalab 250Xi, Thermo Fisher Scientific, Massachusetts, MA, USA) by using Al Kα radiation (1486.6 eV). All binding energies were calibrated using the C1s peak (BE = 284.8 eV) as a standard. Morphology and microstructure were determined by transmission electron microscopy (TEM, Tecnai G2 F30 S-TWIN 200 KV, Hillsboro, OR, USA), high resolution TEM (HRTEM) and selective area electron diffraction (SAED). Measurement of electron paramagnetic resonance (EPR) spectra was performed at 300 K using a Bruker A200 EPR spectrometer (Bruker, Karlsruhe, Germany). The samples were subjected to UV-Vis testing using Shimadzu UV3600 (Shimadzu, Tokyo, Japan). Brunauer–Emmett–Teller (BET) surface area, pore volumes and pore diameter of the samples were determined by using a Micromeritics ASAP 2020C surface area and porosity analyzer (Micromeritics ASAP 2020 BET apparatus, Atlanta, Georgia, GA, USA). The samples were degassed for 4 h at 200 degrees and then analyzed using N_2_ adsorption-desorption in liquid nitrogen.

### 2.3. Photocatalytic Activity Measurements

Photocatalytic performance was investigated by using a 300 W Xeon lamp equipped with a 400 nm cut-off filter to degrade methyl orange (MO), rhodamine B (Rho B) and MB under sunlight and visible light. Water cooled the reactor, to keep the temperature at 25 °C. To put it simply, 50 mg photocatalyst was mixed with 100 mL 10 mg·L^−1^ organic dye solution, and the mixed solution was placed 20 cm away from the photosource. Meanwhile, the mixture was stirred stably. The sample solution was then collected every 30 min of irradiation. The solution collected was centrifuged at 10,000 rpm for 2 min, the mixed solution was separated, and the supernatant was extracted for ultraviolet testing. The dye solutions collected (MO, Rho B and MB) were subjected to absorbance detection using UV-Vis (TU-1900). The degradation efficiencies of MO, Rho B and MB were analyzed by changes in absorption peaks at 462, 554 and 660 nm. The degradation rates of MO, Rho B and MB are calculated by the formula ln (C_0_/C) = kt, where k is the reaction rate constant, C_0_ and C are the initial concentration of the dye and the concentration after the reaction time t. 

### 2.4. Apparent Quantum Efficiency Measurement

The photocatalytic H_2_-production experiments were performed in a 300 mL sealed jacket beaker at ambient temperature and atmospheric pressure. In a typical photocatalytic experiment, 100 mg of Ar-TiO_2_ composite photocatalyst was suspended in 100 mL of aqueous solution containing methanol (20.0 V%) as sacrificial agents for trapping holes. Proper amount of H_2_PtCl_6_ aqueous solution was added in the above solution. Therefore, 3.0 wt% Pt, as a co-catalyst, was in-suit reduced during the photocatalytic hydrogen evolution reaction. Then, evacuation was performed with a vacuum pump to ensure that the reactor was under vacuum. A continuous magnetic stirrer was applied at the bottom of the reactor in order to keep the photocatalyst particles in suspension status during the whole experiment. After 0.5 h of irradiation, the chromatographic inlet was opened, and hydrogen was analyzed by gas chromatograph (GC7806, Beijing Shiwei Spectrum Analysis Instrument Co., Ltd., Beijing, China, TCD, with nitrogen as a carrier gas and 5 A molecular sieve column). All glassware was carefully rinsed with deionized water prior to use. The apparent quantum efficiency (AQE) was measured under the same photocatalytic reaction conditions. A Xe lamp source (300 W, 385 nm) was placed 10 cm directly above the reactor to serve as a light source to initiate a photocatalytic reaction. The optical power density of the Xe lamp source is measured by a strong optical power meter (CEL-NP2000) and the focused area on the beaker is 30 cm^2^. We measured and calculated the number of photons according to Equation (1):
(1)$$\begin{array}{ll} number_{photons} & =\frac{Focused\;intensity\;\times \;Focused\;area}{hv}\\ & =\frac{Focused\;intensity\;\times \;Focused\;area}{h\frac{c}{\lambda }} \end{array}$$

We measured and calculated apparent quantum efficiencies (AQE) according to Equation (2): [50]
(2)$$AQE\left(\%\right)=\frac{number\;of\;reacted\;electrons\;}{number\;of\;incident\;photons}\times 100\;=\frac{{\mathrm{number}\;\mathrm{of}\;\mathrm{evolve}\;H}_{2}\;\mathrm{molecules}\times 2}{\mathrm{number}\;\mathrm{of}\;\mathrm{incident}\;\mathrm{photons}}\times 100$$
where h is the Planck constant (J·s), c is Speed of light (m/s) *and* λ is the wavelength of linght (nm).

## 3. Results and Discussion 

### 3.1. Physicochemical Properties of Catalysts

Figure 1a shows the XRD pattern of the original TiO_2_ and Ar-TiO_2_. It can be seen from the XRD pattern that only the diffraction peak of anatase has no other impurity peaks detected, indicating that the sample contains only anatase. Two samples showed similar peak positions, which were attributed to TiO_2_ (anatase, PDF#21-1272). The diffraction peaks: 2θ = 25°, 37.5°, 48.4°, 53.5°, 55.3° and 62.2° were assigned to the (101), (004), (200), (211) and (204) lattice planes. The XRD pattern of the modified TiO_2_ has a higher strength than the original TiO_2_, which indicated that the Ar plasma caused an improvement in the crystallinity of the modified TiO_2_. The broadened diffraction peaks indicated that the size of the nanocrystals is small [51]. There was no significant difference between the two samples. 

Figure 1b shows the Raman spectrum of the original TiO_2_ and Ar-TiO_2_ in Ar atmosphere. The original sample peaks were: 146 cm^−1^ (E_g_), 196 cm^−1^ (E_g_), 399 cm^−1^ (B_1g_), 514 cm^−1^ (A_1g_ or B_1g_) and 640 cm^−1^ (A_1g_) respectively typical Anatase phase [52,53,54]. The treated sample showed a scattering peak only at 146 cm^−1^ (E_g_). The peak intensity and width of TiO_2_ treated with plasma in Ar atmosphere were lower and wider than those of the original TiO_2_. This indicated that there were some defects on the surface of TiO_2_ after plasma treatment [55]. The Raman scattering selection rule was broken, indicating the surface characteristics of the disordered phase [56]. Raman analysis showed that the molecular vibration of the surface crystallinity of TiO_2_ was seriously affected by the plasma treatment. 

The FTIR spectra of the original and Ar-TiO_2_ are shown in Figure 1c. An absorption peak appearing near 523 cm^−1^ was due to the Ti-O-Ti bond of nano-titanium dioxide, and the absorption peaks at 3500–3000 cm^−1^ and 1630 cm^−1^ were caused by the tensile vibration of the hydroxyl group. The hydroxyl group was adsorbed on the surface of the material [57]. The C-O band (1437 cm^−1^) was detected due to the formation of CO_2_ on the surface of TiO_2_ [58,59]. 

X-ray photoelectron spectroscopy (XPS) demonstrates the defects and the chemical composition of Ar-TiO_2_. The XPS spectra of the elements O 1s and Ti 2p are shown in Figure 2b,c. In Figure 2b, O 1s was divided into three peaks, 530.0, 531.6 and 533.1 eV, respectively, which corresponds with lattice oxygen (O_latt_), defect oxygen (O_def_) and surface oxygen (O_surf_) [60,61]. 

Figure 2b shows that, compared with the original TiO_2_, the O_def_ content in the Ar-TiO_2_ is significantly increased, indicating that the plasma effectively removed oxygen, and thus some oxygen vacancies are produced in the crystal lattice of TiO_2_. In calculation, the defect-oxygen content of the plasma-treated TiO_2_ was four times that of the original TiO_2_ (Table 1) [60]. 

Figure 2c presents the Ti 2p photoelectron spectra of TiO_2_, the Ti 2p peaks at 464.76, 463.7, 459.03, and 458.48 eV were observed corresponding to Ti^4+^ 2p_1/2_, Ti^3+^ 2p_1/2_, Ti^4+^ 2p_3/2_, and Ti^3+^ 2p_3/2_, respectively [55,62]. However, it can be seen that after the plasma treatment in the Ar atmosphere, the peaks of Ti^4+^ at 459.03 eV and 464.76 eV were shifted by 0.36 eV and 0.24 eV, and the content of Ti^3+^ peak was increased, implying that Ti^3+^ was generated as a result of the Ti^4+^ reduction. A large amount of Ti^3+^ and oxygen vacancy in Ar-TiO_2_ could make the coordination number of Ti-O-Ti and the surface lattice structure change, thus generating the defects [63]. According to the results of XPS analysis, the plasma-treated titanium dioxide had abundant defects and oxygen vacancies, which played a key role in the degradation of the photocatalyst.

Electron paramagnetic resonance (EPR) was used to further demonstrated oxygen vacancies and Ti^3+^ defects of the samples. The magnetization results recorded at 300 K clearly showed surface oxygen vacancies and unpaired spins electrons of Ti^3+^ 3d1 in the samples. Figure 2d shows the EPR results of the original TiO_2_ and Ar-TiO_2_. The index values of the g peaks at 1.95 and 2.004 corresponded to Ti^3+^ and oxygen vacancies in the TiO_2_ lattice, respectively [60,64,65]. The content of Ti^3+^ and oxygen vacancies in plasma treatment were significantly higher than that of the original TiO_2_, which was consistent with the analysis results of the XPS spectrum.

In order to test the optical performance of the photocatalyst, UV-Vis among 200 nm and 800 nm was surveyed. As shown in Figure 3a, all of the TiO_2_ photocatalysts showed the significant absorption of light in the ultraviolet region (<400 nm) and the light absorption of the Ar-TiO_2_ photocatalyst moved toward the visible region. The band gap of each sample was estimated by the simulation calculation, TiO_2_ was 3.21 eV and the Ar-TiO_2_ was 3.17 eV (inset of Figure 3a). This value is the effect of the extrapolation to zero (αhν) ^1/2^ curve on photon energy, where α is the absorption factor and hν is the light quantum energy [53,56]. Based on the XPS valence band (VB) spectrum (Figure 3b), the valence band position of Ar-TiO_2_ was 2.93 eV, which was lower than the original TiO_2_ (3.02 eV). These results indicated that the oxygen vacancies (or Ti^3+^ species) were produced during the Ar gas plasma treatment and caused the band gap of the Ar-TiO_2_ photocatalyst to be narrowed. The electronic band-gap structure and photocatalytic degradation mechanism of TiO_2_ samples were shown in Figure 4.

### 3.2. Morphological Characterization

In order to eliminate the impact of catalyst adsorption on photocatalytic performance, BET tests were carried out on the two materials. The N_2_ adsorption-desorption isotherms and Barrett-Joyner-Halenda (BJH) pore size distribution curves of TiO_2_ and Ar-TiO_2_ catalysts are shown in Figure 1d. Both catalyst isotherms are typical type IV isotherms, H1 type hysteresis loops, indicating that the International Union of Pure and Applied Chemistry (IUPAC) classification are a mesoporous material [66]. The BET surface areas of TiO_2_ and Ar-TiO_2_ were about 124.7 m^2^/g and 121.3 m^2^/g, respectively (Table 2), indicating that the specific surface areas of the two catalysts were not much different. 

The pore size distribution (Figure 1d) was determined by the BJH method from the desorption branch of the isotherm, indicating that these TiO_2_ nanoparticles have a very pronounced mesoporous structure. The average pore diameters of TiO_2_ and Ar-TiO_2_ were 5.6 and 5.6 nm, respectively (Table 2). The mesoporous size distribution of the TiO_2_ nanoparticles and the mesoporous size distribution of the plasma-treated TiO_2_ nanoparticles were not greatly different, indicating that the pore size uniformity of the two materials were near same. The effect of physical adsorption on photocatalytic performance can be ruled out.

TEM and HRTEM images of TiO_2_ and Ar-TiO_2_ are shown in Figure 5. Figure 5a,d shown that there were no significant changes in the morphology of the TiO_2_ after the treatment, and both of them exhibit irregular large spherical nanoparticles. These irregular spherical nanoparticles consist of a plurality of small particles. As shown in Figure 5c, the TiO_2_ catalyst showed clear lattice fringes. The lattice spacing values were 0.352, 0.2378 and 0.189 nm, corresponding to the (101), (004) and (200) crystal faces of TiO_2_, respectively. However, in Figure 5f, after plasma treatment (Ar-TiO_2_), the surface of TiO_2_ changed significantly, localized regions were destroyed and uneven, lattice defects were generated, and many deformations occurred. The lattice can increase the amorphous crystal structure and oxygen vacancies [56]. Therefore, defects on the surface of the catalyst after plasma treatment expose more active sites, improving photocatalytic activity.

### 3.3. Photocatalytic Performance

The photocatalytic degradation experiments of MO, Rho B and MB dye solutions in TiO_2_ and Ar-TiO_2_ photocatalysts under sunlight and visible light (λ ≥ 400 nm) were estimated. In a typical photocatalytic reaction, TiO_2_ absorbs energy greater than its forbidden band width and excites electron-hole pairs in the conduction and valence bands. Then these segregated charges transfer to the catalyst surface to take part in the following photocatalytic degradation reaction. Above all, the strong active O species (O_2_^−·^ and ·OH) produced in the photocatalytic process have superior redox properties and are used to degrade organic pollutants in water, resulting in an effective photocatalytic degradation reaction.

Figure 6a shows that the Ar-TiO_2_ (99.6% efficiency) photocatalyst achieved better MO degradation performance than TiO_2_ (49% efficiency) after 30 min of sunlight exposure. Degradation rate k (kt = ln(C_0_/C)), the degradation rate of the TiO_2_ sample was 0.66 h^−1^, and the degradation rate of Ar-TiO_2_ sample was 5.38 h^−1^. Similarly, Ar-TiO_2_ photocatalyst showed excellent performance in the degradation of MO solution after visible light (λ ≥ 400 nm) for 150 min (Figure 6b). The degradation rate of TiO_2_ was 0.26 h^−1^, and the degradation rate of Ar-TiO_2_ was 2.14 h^−1^. As shown in Figure 6c–f, similar results were obtained for additional degradation experiment of Rho B and MB under sunlight and visible light (λ ≥ 400 nm) illumination. The degradation rate after dye degradation is shown in Table 3. 

Figure 6c,e show that the Ar-TiO_2_ sample has excellent removal performance in the degradation test of Rho B and MB solution under 120 min of sunlight, while the degradation efficiency of the TiO_2_ sample was relatively lower. Under sunlight, the degradation efficiency of Ar-TiO_2_ to MO was higher than that of Rho B and MB. Therefore, the photodegradation properties of the Ar-TiO_2_ sample depend on the dye used, MO (anionic dye), Rho B and MB (cationic dye). This was because the degradation efficiency of the catalyst is impacted by the pH value of the organic dye solution. The pH changed the surface electric charge of TiO_2_, which affected the adsorption of the organic dye on the surface of TiO_2_, and influenced the rate on reaction and changed the degradation performance of the photocatalyst [67]. The same test on photocatalyst under visible light (λ ≥ 400 nm) was performed. It can be seen from the Figure 6b,d,f that Ar-TiO_2_ catalyst exhibited excellent degradation performance in degrading MO, Rho B and MB under 150 min visible light irradiation. The excellent photocatalytic performance of the Ar-TiO_2_ sample was attributed to the reduction of the TiO_2_ band gap by plasma treatment, resulting in an expansion of the optical absorption range from the ultraviolet to the visible region, and an increase in the amount of active oxygen and Ti^3+^ defects. 

The apparent quantum efficiencies (AQE) of the TiO_2_ and Ar-TiO_2_ catalysts was calculated by photocatalytic hydrogen production experiments under full-spectrum conditions. As shown in Table 4, the optical power density was 800 mW·cm^−2^ by the strong optical power meter measured. The number of incident photons were 4.615 × 10^19^ calculated by the Formula (1) calculated. As can be seen from Table 4, the sample of TiO_2_ exhibited a H_2_ evolution rates (HER) of 320 μmol·g^−1^·h^−1^ with a quantum efficiency of 41.7 % under UV light irradiation (λ = 385 nm) after 2.5 h. Compared with TiO_2_, the sample of Ar-TiO_2_ exhibited the highest H_2_ evolution rates (530 μmol·g^−1^·h^−1^) with a quantum efficiency of 69.0 %. It can be concluded Ar-TiO_2_ has higher light utilization efficiency.

## 4. Conclusions

In conclusion, low temperature DBD plasma was successfully employed to treat the surface of TiO_2_ nanoparticles in Ar atmosphere. The Ar-TiO_2_ obtained was used for photocatalytic degradation of organic dyes. The results indicated that a large number of oxygen vacancies and Ti^3+^ defects were subsequently generated to promote the decomposition of the reactant molecules, thereby improving the reaction efficiency. Plasma treatment shortened the width of the TiO_2_ band gap and expanded the light absorption range. The photocatalytic performance indicated that the plasma-treated TiO_2_ was much better than the original TiO_2_ nanoparticles in photocatalytic degradation of organic dyes under sunlight. Therefore, plasma can be an effective means to optimize photocatalytic degradation of TiO_2_ nanoparticles.

## Figures and Tables

**Figure 1 nanomaterials-09-00720-f001:**
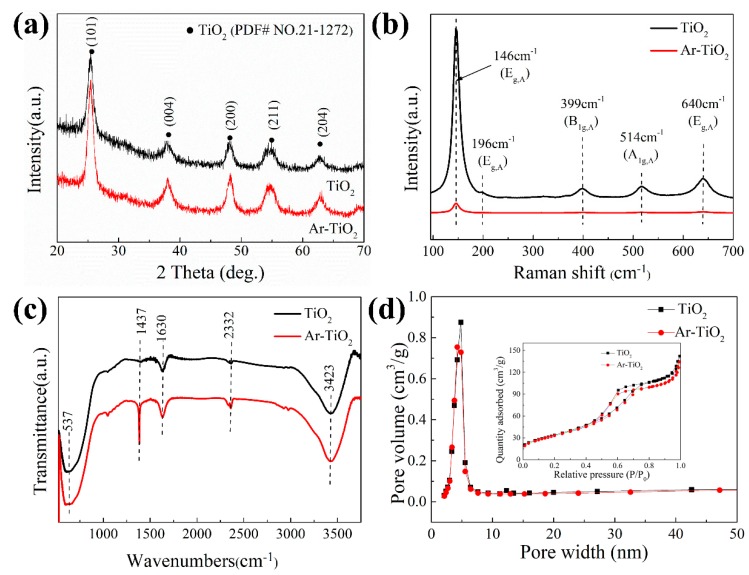
Spectra of (**a**) X-ray diffractometer (XRD) spectra, (**b**) Raman spectra, (**c**) Fourier transforms infrared spectra (FTIR) spectra and (**d**) BET spectra for the Ar-TiO_2_ and pristine TiO_2_ catalysts. Insert: the N_2_ adsorption-desorption isotherms.

**Figure 2 nanomaterials-09-00720-f002:**
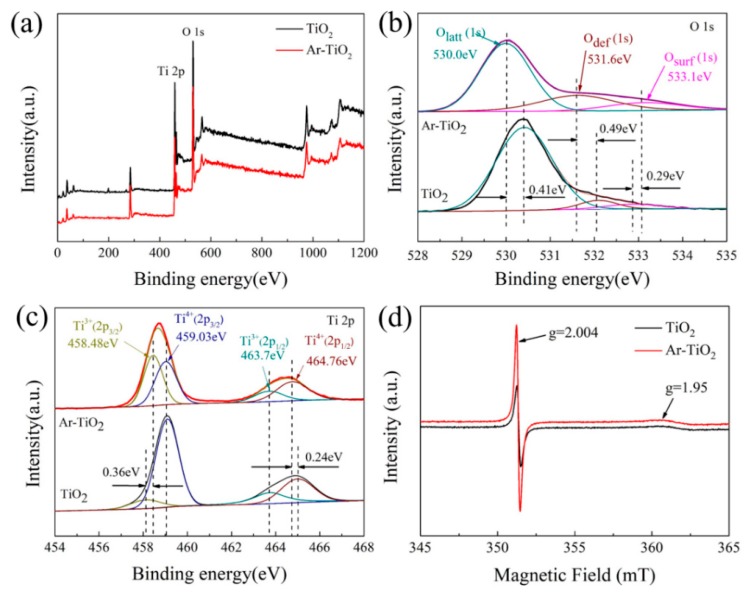
X-ray photoelectron spectroscopy (XPS) spectra of (**a**) survey spectra, (**b**) O 1s, (**c**) Ti 2p and (**d**) Electron paramagnetic resonance (EPR) spectra for the Ar-TiO_2_ and pristine TiO_2_ catalysts.

**Figure 3 nanomaterials-09-00720-f003:**
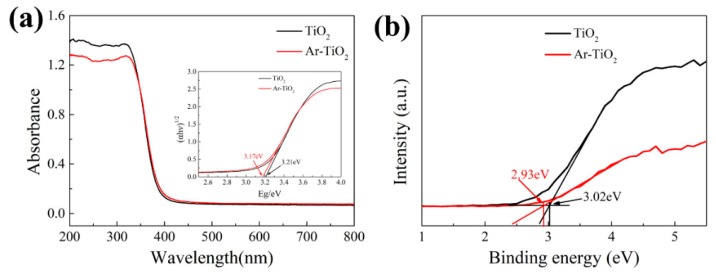
(**a**) Ultraviolet (UV)-visible absorption spectra DRS, the insets in Figure 3 (**a**) is the corresponding plots of transformed Kubelka-Munk function versus the energy of photon, (**b**) XPS valence band spectra.

**Figure 4 nanomaterials-09-00720-f004:**
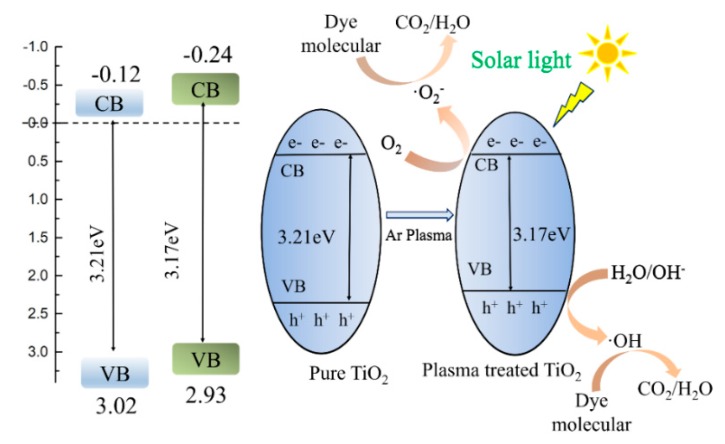
Schematic drawing illustrating the mechanism of charge separation and photocatalytic activity of the TiO_2_ photocatalyst under solar light irradiation.

**Figure 5 nanomaterials-09-00720-f005:**
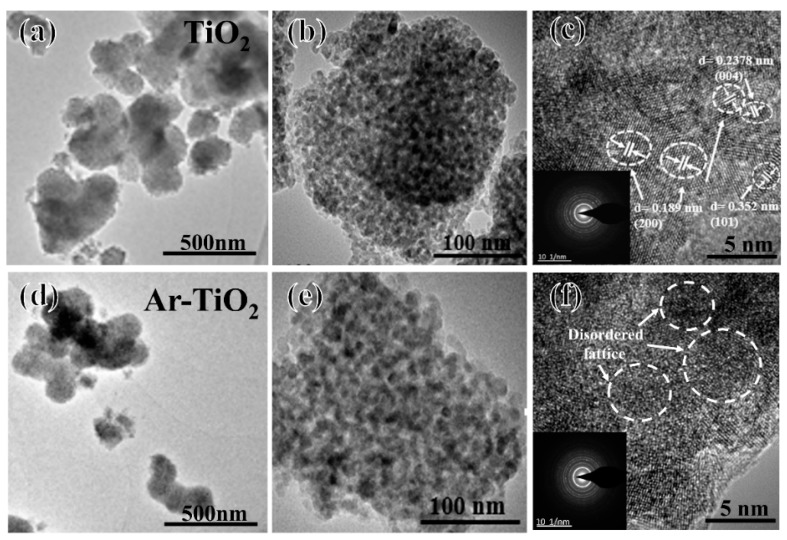
The results of transmission electron microscopy (TEM) of (**a**,**b**) TiO_2_ and (**d**,**e**) Ar-TiO_2_. High resolution TEM (HRTEM) image curves of (**c**) TiO_2_ (inset: SAED) and (**f**) Ar-TiO_2_ (insert: SAED).

**Figure 6 nanomaterials-09-00720-f006:**
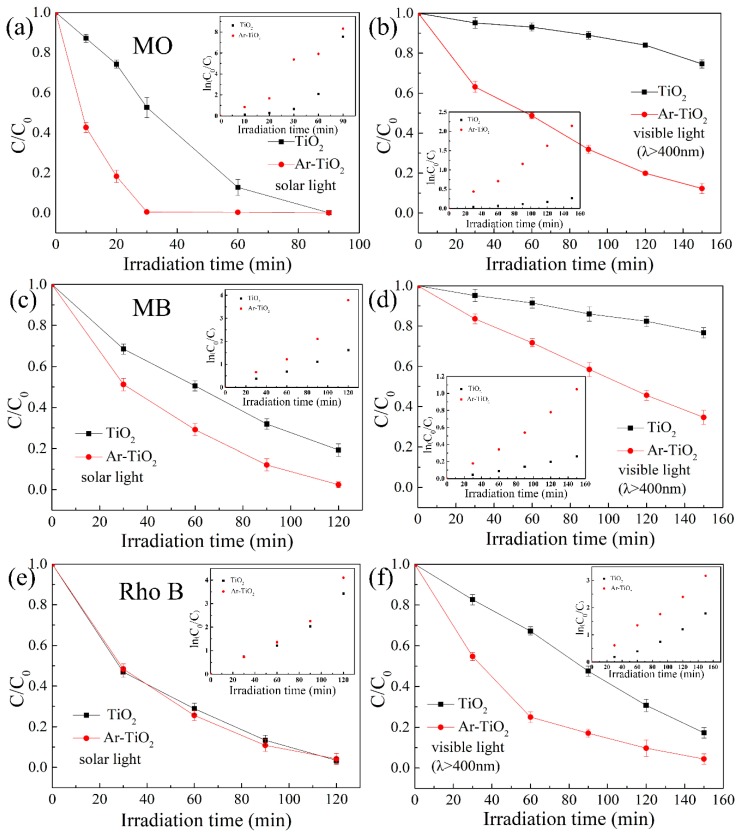
Removal of (**a**,**b**) methyl orange (MO), (**c**,**d**) methylene blue (MB) and (**e**,**f**) rhodamine B (Rho B) by TiO_2_ and Ar-TiO_2_ (inset: corresponding degradation rate images) under solar light and/or visible light (λ ≥ 400 nm) irradiation. Solution concentration: 10 mg/L; organic solution: 100 mL; catalyst: 50 mg.

**Table 1 nanomaterials-09-00720-t001:** Surface atomic species on the surface of TiO_2_ and Ar-TiO_2_ catalysts.

Samples	Surface Atomic Concentration (%)	
	Ti^3+^/Ti	O_def_/(O_latt_ + O_def_ + O_surf_)
TiO_2_	20.9	6.0
Ar-TiO_2_	45.2	24.4

**Table 2 nanomaterials-09-00720-t002:** BET specific surface are, pore volume and pore size of TiO_2_ and Ar-TiO_2_ catalysts, respectively.

Sample	Surface Area (m^2^/g)	Pore Volume (cm^3^/g)	Pore Size (nm)
TiO_2_	124.7	0.23	5.6
Ar-TiO_2_	121.3	0.21	5.6

**Table 3 nanomaterials-09-00720-t003:** Degradation rates, k (h^−1^), of TiO_2_ and Ar-TiO_2_ after photocatalytic degradation of MO, MB and Rho B dyes.

		TiO_2_	Ar-TiO_2_
MO	solar light	7.55	8.32
	visible light (λ ≥ 400 nm)	0.26	2.14
MB	solar light	1.62	3.78
	visible light (λ ≥ 400 nm)	0.26	1.05
Rho B	solar light	3.42	4.10
	visible light (λ ≥ 400 nm)	1.79	3.17

**Table 4 nanomaterials-09-00720-t004:** Apparent quantum efficiencies (AQE) and H_2_ evolution rates (HER) of the TiO_2_ and Ar-TiO_2_ catalysts at optical power density of 800 mW·cm^−2^ with incident photon number of 4.615 × 10^19^.

Samples	Time (h)	1	1.5	2	2.5
TiO_2_	HER (μmol·g^−1^·h^−1^)	98	287	306	320
	AQE (%)	8.7	35.2	39.8	47.7
Ar-TiO_2_	HER (μmol·g^−1^·h^−1^)	240	506	524	530
	AQE (%)	31.3	66.0	68.1	69

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
