# Peer review of "Enhanced Photocatalytic Degradation of Organic Dyes via Defect-Rich TiO_2_ Prepared by Dielectric Barrier Discharge Plasma"

_nanomaterials, 2019, doi:10.3390/nano9050720_

Round 1
Reviewer 1 Report
The manuscript deals with the preparation of Ar-plasma treated Tio2 nanoparticles for the degradation of organic dyes. I appreciate authors for their thought and theme of work. The preparation and characterization results are well written. But, there are many reports in the literature for the degradation of organic dyes using TiO2 and its hybrid materials which are shows better degradation as well as economically feasible materials. Hence, The authors have missed the explanation of novelty, economic feasibility and degradation mechanism of dyes using present material as compared other reported materials.
1. What are degradation mechanisms involved in different light sources(UV and Solar energy)? explain.
2. Add error bars on degradation data reported in Fig.5.
3. The obtained degradation results evaluate in terms of statistical analysis and compare with reported one literature.
4. The abstract seems to be part of the introduction, its need to revise by including significant findings.
5. The conclusions should be improved.
Author Response
1. What are degradation mechanisms involved in different light sources(UV and Solar energy)? explain.
Reply: Thank you very much for the reviewer’s helpful comments. In photocatalysis, light of energy which is greater than the band gap of TiO2 excites an electron to the conduction band and generates a positive hole in the valence band. Holes in the valence band can oxidize -OH (or water) at the surface for production of·OH radicals and electrons in the conduction band can be rapidly trapped by molecular oxygen adsorbed on the TiO2, which is reduced to form superoxide radical anion, O2-· reacting with H+ to create hydroperoxyl radical (·OOH). Usually, the advancement of the photocatalytic activities of TiO2 depends on the surface bonds providing enhanced optical absorption and an effective separation of photogenerated electron-hole pairs. The hydroxyl radical and the superoxy radical generated by the electron-hole pair have redox capability, and the organic dye can be degraded into H2O and CO2 by a redox reaction. [1-3] Under sunlight and ultraviolet light, the degradation mechanism of TiO2 is the same, except that the energy of ultraviolet light is stronger than that of sunlight, which easily excites electrons and generates electron-hole pairs, thus producing more hydroxyl radicals and superoxides. Free radicals, therefore easily degraded under ultraviolet light than under the sun.
[1] An, H.R.; Yong, C.H.; Kim, H.; Jin, Y.H.; Park, E.C.; Park, S.Y.; Jeong, Y.; Park, J.I.; Kim, J.P.; Lee, Y.C. Studies on mass production and highly solar light photocatalytic properties of gray hydrogenated-TiO2 sphere photocatalysts. J. Hazard. Mater. 2018, 358, 222-233.
[2] Kong X , Xu Y , Cui Z , et al. Defect enhances photocatalytic activity of ultrathin TiO2 (B) nanosheets for hydrogen production by plasma engraving method. Applied Catalysis B: Environmental, 2018, 230, 11-17.
[3] Hu, M.; Xing, Z.; Yan, C.; Li, Z.; Xu, Y.; Xiu, Z.; Zhao, T.; Yang, S.; Wei, Z., Ti3+ self-doped mesoporous black TiO2 /SiO2 /g-C3N4 sheets heterojunctions as remarkable visible-lightdriven photocatalysts. Appl. Catal. B: Environ. 2018, 226, 499–508.
2. Add error bars on degradation data reported in Fig.5.
Reply: Thank you very much for the reviewer’s helpful comments. We have added error bars on degradation data reported in Fig.6.
3. The obtained degradation results evaluate in terms of statistical analysis and compare with reported one literature.
Reply: Thank you very much for the reviewer’s helpful comments. As shown in Table R1, compared to other work, the catalysts we prepared have excellent degradation properties in the degradation of MO, MB, and Rho B dyes.
[1] Lu L, Shan R, Shi Y, et al. Novel TiO2/biochar composite catalysts for photocatalytic degradation of methyl orange. Chemosphere, 2019, 222, 391-398.
[2] Fu C, Gong Y, Wu Y, et al. Photocatalytic Enhancement of TiO2 by B and Or Co-Doping and Modulation of Microstructure. Applied Surface Science, 2016, 379, 83-90.
[3] Xin, Y.; Zhao, X.; Li, Y.; Zhao, Q.; Zhou, X.; Yuan, Q., CTAB-assisted synthesis of mesoporous F–N-cooped TiO2 powders with high visible-light-driven catalytic activity and adsorption capacity. Journal of Solid State Chemistry 2008, 181, 1936-1942.
[4] Tian J, Shao Q, Zhao J, et al. Microwave solvothermal carboxymethyl chitosan templated synthesis of TiO2/ZrO2 composites toward enhanced photocatalytic degradation of Rhodamine B. Journal of colloid and interface science, 2019, 541, 18-29.
4. The abstract seems to be part of the introduction, its need to revise by including significant findings.
Reply: Thank you very much for the reviewer’s helpful comments. We made the following revisions:
The dye wastewater produced in the printing and dyeing industry takes a serious harm to the natural environment. Usually, when used TiO2 nanoparticals as a photocatalyst, it absorbs only ultraviolet light, and several approaches have been taken to narrow the band gap of TiO2. Thus, we demonstrated a facile and environmental friendly method to enhancing degradation of organic dyes by introducing defects of oxygen vacancy and Ti3+ in surface TiO2 nanoparticals through the plasma engravin treatment. A large number of oxygen vacancies and Ti3+ defects were generated on the surface of the TiO2 nanoparticles via Ar plasma. Compared with pristine TiO2 nanoparticles, the as-obtained Ar plasma-treated TiO2 (Ar-TiO2) nanoparticles make energy band gap reduced from 3.21eV to 3.17eV and exhibit enhanced photocatalytic degradation of organic dyes. The obtained Ar-TiO2 exhibited excellent degradation properties of MO, the degradation rate under sunlight irradiation was 99.6% in 30 minutes, and the photocatalytic performance was about twice that of the original TiO2 nanoparticles (49%). The degradation rate under visible light (λ> 400nm) irradiation was 89% in 150 minutes, and the photocatalytic performance of the Ar-TiO2 was approaching ~4 times higher than that of the original TiO2 nanoparticles (23%). Ar-TiO2 also showed good degradation performance in degrading Rho B and MB. In this work, an attempt is made to improve the photocatalytic activity of other metal oxides, and provide a new method for improving the photocatalytic activity of other metal oxides.
5. The conclusions should be improved.
Reply: Thank you very much for the reviewer’s helpful comments. We made the following revisions:
In conclusion, low temperature dielectric barrier discharge plasma was successfully employed to treatment the surface of TiO
In conclusion, low temperature dielectric barrier discharge plasma was successfully employed to treatment the surface of TiO2 nanoparticles in Ar aumosphere. The obtained Ar-TiO2 was used for photocatalytic degradation of organic dyes. The results indicate that A large number of oxygen vacancies and Ti3+ defects are subsequently generated to promote the decomposition of the reactant molecules, thereby improving the reaction efficiency. Plasma treatment shortens the width of the TiO2 band gap and expands the light absorption range. The photocatalytic performance indicates that the plasma-treated TiO2 is much batter than the original TiO2 nanoparticles in photocatalytic degradation of organic dyes under sunlight. Therefore, plasma can be an effective means to optimize photocatalytic degradation of TiO2 nanoparticles.

Reviewer 2 Report
The authors report on the photocatalytic activity of Ar plasma treated TiO2 nanoparticles on dye degradation. Considerable enhancement of dye photodegradation was observed for the plasma modified TO2 NPs that was related to the presence of oxygen vacancies and Ti3+ defects and the small bandgap variation. Overall, the work is well aimed and the materials sufficiently characterized but there are several points that need careful consideration, especially in the materials’ photocatalytic evaluation:
1) The reported band gap reduction by ca.0.04 eV is rather small to extend significantly light absorption and play any significant role in the visible light photocatalytic activity. This point should be revised.
2) How is the visible light dye degradation, especially in the case of rhodamine B, using the untreated TiO2 NPs explained? Moreover, the degradation rate of the anionic methyl orange azo-dye is considerably higher than the cationic MB dye that usually adsorbs preferentially at the titania surface and is easily degraded . How is this explained? The dark adsorption of the dye molecules on the titania NPs should be reported so that variations at the starting dye concentration when light is turned on are taken into account. Blank tests with the dye self-degradation without catalyst under the same irradiation conditions should be also performed and reported.
3) In order to validate the photocatalytic performance of the prepared TiO2 NPs, an independent test should be provided by photocatalytic experiments using the benchmark P25 powder catalyst under the same reaction conditions (catalyst loading, dye concentration, irradiation etc).
4) The lnC0/C vs t plots should be presented by scatter data in the insets of Fig. 5 and the corresponding apparent constants kapp should be derived by linear fits. The irradiation power density by the Xe lamp should be reported.
5) The English should be checked throughout the manuscript and typos should be corrected. Loose or incorrect terms like the following should be amended:
l. 40 replace “cracking of water” by water splitting.
l. 48 replace “dope” with another word like “modify” or “couple” (there is no doping of TiO2 with other semiconducting oxides).
l. 127 Please correct the phrase “in crystal the structure and crystal the phase,”
l. 131 The “improvement in the crystallinity” is in contrast to the generation of defects and the excessive broadening of the Raman spectra after plasma treatment.
l.137 There are no “diffraction peaks” in the Raman spectra
l.138 Strong statements like “destruction of the original symmetry of the TiO2 lattice after plasma treatment” are hardly justified by the data e.g. the XRD patterns and shoudl be removed.
Author Response
Reviews 2
The authors report on the photocatalytic activity of Ar plasma treated TiO2 nanoparticles on dye degradation. Considerable enhancement of dye photodegradation was observed for the plasma modified TO2 NPs that was related to the presence of oxygen vacancies and Ti3+ defects and the small bandgap variation. Overall, the work is well aimed and the materials sufficiently characterized but there are several points that need careful consideration, especially in the materials’ photocatalytic evaluation:
1) The reported band gap reduction by ca.0.04 eV is rather small to extend significantly light absorption and play any significant role in the visible light photocatalytic activity. This point should be revised.
Figure 2. XPS spectra of (a) survey spectra, (b) O 1s, (c) Ti 2p, (d) EPR spectra, for the Plasma treated and pristine TiO2 catalysts.
Reply: Thank you very much for the reviewer’s helpful comments. The reduction in the forbidden band width enhances the light absorbing ability, but the photocatalytic activity does not depend entirely on the reduction of the band gap. It can be seen from the XPS diagrams of Ti and O, many defects of Ti3+ and oxygen vacancies in the plasma treated TiO2 is able to change the coordination number of Ti-O-Ti surface lattice structure, and further form andefect state. Compared with pristine TiO2, Ti3+ and vacancy oxygen in plasma treated TiO2 lead to a relative low electronegativity and high polarizability [1]. The Ti3+ defects are gradually accumulated in the sample and usually accompanied with losing of oxygen[2]. Therefore, there were more super oxygen vacancies to be detected in the plasma treated TiO2 as shown in Fig. 2(d). With the photocatalytic reactions going on, a lot of excited-state electrons are generated to form O2- with dissolved oxygen. Under light irradiation, these active species (e.g. O2-, OH·, or h+) can promote the separation and migration of photoelectrons, and indirectly improve the utilization of photoelectrons during the water splitting [3, 4], thereby improving the degradation efficiency.
[1] T. Lin, C. Yang, Z. Wang, H. Yin, X. Lü, F. Huang, J. Lin, X. Xie, M. Jiang, Energ. Environ. Sci. 7 (2014) 967-972
[2] Y. Yang, G. Liu, J.T. Irvine, H.M. Cheng, Adv. Mater. 28 (2016) 5850-5856.
[3] L. Yu, Y. Shao, D. Li, Appl. Catal. B: Environ. 204 (2017) 216-223.
[4] R. Fu, S. Gao, H. Xu, Q. Wang, Z. Wang, B. Huang, Y. Dai, RSC Adv. 4 (2014) 37061-37069.
2)How is the visible light dye degradation, especially in the case of rhodamine B, using the untreated TiO2 NPs explained? Moreover, the degradation rate of the anionic methyl orange azo-dye is considerably higher than the cationic MB dye that usually adsorbs preferentially at the titania surface and is easily degraded. How is this explained? The dark adsorption of the dye molecules on the titania NPs should be reported so that variations at the starting dye concentration when light is turned on are taken into account. Blank tests with the dye self-degradation without catalyst under the same irradiation conditions should be also performed and reported.
Figure 2. XPS spectra of (a) survey spectra, (b) O 1s, (c) Ti 2p, (d) EPR spectra, for the Plasma treated and pristine TiO2 catalysts.
1. Reply: Thank you very much for the reviewer’s helpful comments. Untreated TiO2 was prepared by hydrolyzing titanium source. It can be seen from XPS and EPR charts that the surface of untreated TiO2 has a small amount of Ti defects and oxygen vacancies. These small defects cause TiO2 to produce active substances with redox ability. (e.g. O2-, OH·, or h+), thus having good degradation ability.[1, 2]
[1] An, H.R.; Yong, C.H.; Kim, H.; Jin, Y.H.; Park, E.C.; Park, S.Y.; Jeong, Y.; Park, J.I.; Kim, J.P.; Lee, Y.C. Studies on mass production and highly solar light photocatalytic properties of gray hydrogenated-TiO2 sphere photocatalysts. J. Hazard. Mater. 2018, 358, 222-233.
[2] Kong X , Xu Y , Cui Z , et al. Defect enhances photocatalytic activity of ultrathin TiO2, (B) nanosheets for hydrogen production by plasma engraving method[J]. Applied Catalysis B: Environmental, 2018, 230, 11-17.
2. Reply:Thank you very much for the reviewer’s helpful comments. The Photodegradation of dyes are affected by the pH of the solution. The variation of solution pH changes the surface charge of TiO2 particles and shifts the potentials of catalytic reactions. As a result, the adsorption of dye on the surface is altered thereby causing a change in the reaction rate. Under acidic or alkaline condition the surface of Titania can be protonated or deprotonated respectively according to the following reactions: [1]
TiOH + H+→ TiOH2+ (9)
TiOH + OH- → TiO- + H2O (10)
Thus, the surface of titanium dioxide will remain positively charged in an acidic medium and negatively charged in an alkaline medium. It has been reported that titanium dioxide has higher oxidation activity at lower pH, but excessive H+ will reduce the reaction rate. Due to the weak acidity of anion methyl orange orange-red solution, the surface of TiO2 is positively charged, and TiO2 presents as strong lewis acid. In other words, the anionic dye ACTS as a strong lewis base and can be readily attached to the positively charged catalyst surface. Under weakly alkaline and alkaline conditions, TiO2 has a negative charge on the surface. This complexation process is not conducive to Columbic repulsion, which is probably caused by competitive adsorption of hydroxyl groups and dye molecules and by negatively charged catalysts containing dye molecules.[2]
[1] Davis RJ, Gainer JL, Neal GO, et al. Photocatalytic decolorization of wastewater dyes. Water Environ Res.
1994; 66(1):50–53.
[2] Mozia S, Morawski AW, Toyoda M, et al. Application of anatase-phase TiO2 for decomposition of azo dye in a photocatalytic membrane reactor. Desalination. 2009; 241(1–3):97–105.
3 Reply: Thank you very much for the reviewer’s helpful comments. As shown in Fig.R1, we performed blank group, dark adsorption, and P25 degradation experiments on the three dyes. The experimental results showed that the three dyes showed little change in the environment without catalyst. Dark adsorption experiments show that the catalyst adsorbs a small amount of organic dye. P25 degradation experiments showed that the degradation efficiency: Ar-TiO2>TiO2>P25. Since commercial P25 is a combination of anatase phase and rutile phase, anatase phase has superior photocatalytic performance compared to rutile phase, and Ar-TiO2 has a wide range of defects and therefore has excellent photocatalytic properties.
Fig. R1 Removal of (a, b) methyl orange (MO), (c,d) methylene blue (MB), and (e,f) rhodamine B (Rho B) by TiO2, Ar-TiO2 and P25 under no catalyst, dark, solar light and/or visible light (λ>400nm) irradiation. Solution concentration: 10 mg/L; organic solution: 100 mL; catalyst: 50mg.
3) In order to validate the photocatalytic performance of the prepared TiO2 NPs, an independent test should be provided by photocatalytic experiments using the benchmark P25 powder catalyst under the same reaction conditions (catalyst loading, dye concentration, irradiation etc).
Reply: Thank you very much for the reviewer’s helpful comments. As shown in Fig.R2, P25 degradation experiments showed that the degradation efficiency: Ar-TiO2>TiO2>P25. Since commercial P25 is a combination of anatase phase and rutile phase, anatase phase has superior photocatalytic performance compared to rutile phase, and Ar-TiO2 has a wide range of defects and therefore has excellent photocatalytic properties.
Fig. R2 Removal of (a, b) methyl orange (MO), (c,d) methylene blue (MB), and (e,f) rhodamine B (Rho B) by TiO2, Ar-TiO2 and P25 under no catalyst, dark, solar light and/or visible light (λ>400nm) irradiation. Solution concentration: 10 mg/L; organic solution: 100 mL; catalyst: 50mg.
4) The lnC0/C vs t plots should be presented by scatter data in the insets of Fig. 5 and the corresponding apparent constants kapp should be derived by linear fits. The irradiation power density by the Xe lamp should be reported.
Reply: Thank you very much for the reviewer’s helpful comments. (1) We have represented the scatter data in the lnC0 / C vs t diagram in the illustration in Fig. 6. The corresponding apparent constants kapp should be derived by linear fits, as in Table R2.
Figure 6. Removal of (a,b) methyl orange (MO), (c,d) methylene blue (MB), and (e,f) rhodamine B (Rho B) by TiO2, and Ar-TiO2 under solar light and/or visible light (λ>400nm) irradiation. Solution concentration: 10 mg/L; organic solution: 100 mL; catalyst: 50mg.
(2) Table R2. Linear fitting of the apparent rate constant kapp(min-1) of TiO2 and Ar-TiO2 after photocatalytic degradation of MO, MB and Rho B dyes.
(3) The irradiation power density of the xenon lamp is 800mW·cm-2.
5) The English should be checked throughout the manuscript and typos should be corrected. Loose or incorrect terms like the following should be amended:
l. 40 replace “cracking of water” by water splitting.
Reply: Thank you very much for the reviewer’s helpful comments. We have modified here as follows: photocatalytic by water splitting to produce hydrogen, carbon dioxide catalytic reduction and antibacterial action.
l. 48 replace “dope” with another word like “modify” or “couple” (there is no doping of TiO2 with other semiconducting oxides).
Reply: Thank you very much for the reviewer’s helpful comments. We have modified here as follows: To inprove the photocatalytic performance of TiO2 some transition metal oxides modifyin TiO2,
l. 127 Please correct the phrase “in crystal the structure and crystal the phase,”
Reply: Thank you very much for the reviewer’s helpful comments. We have modified here as follows: It can be seen from the XRD pattern that only the diffraction peak of anatase has no other impurity peaks detected, indicating that the sample c ontains only anatase.
l. 131 The “improvement in the crystallinity” is in contrast to the generation of defects and the excessive broadening of the Raman spectra after plasma treatment.
Reply: Thank you very much for the reviewer’s helpful comments. XRD characterizes the degree of crystallinity through Xie Le formula, mainly the influence of crystal plane diffraction and peak width. The wider the diffraction peak, the lower the degree of order, the more uneven the crystal plane, the worse the crystallinity; The Raman effect originates from molecular vibration (and lattice vibration) and rotation. The peak intensity and peak width of plasma treated TiO2 is lower and broader than that of pristine TiO2, respectively. This is because the plasma treatment seriously influence molecular vibrations of surface crystallinity and break down the Raman scattering selection rules [1, 2]
[1] J. Li, M. Zhang, Z. Guan, Q. Li, C. He, J. Yang, Appl. Catal. B: Environ. 206 (2017) 300-307.
[2] B. Santara, P.K. Giri, K. Imakita, M. Fujii, Nanoscale 5 (2013) 5476-5488.
l.137 There are no “diffraction peaks” in the Raman spectra
Reply: Thank you very much for the reviewer’s helpful comments. We have modified here as follows: The treated sample showed Scattering peak only at 146 cm-1 (Eg).
l.138 Strong statements like “destruction of the original symmetry of the TiO2 lattice after plasma treatment” are hardly justified by the data e.g. the XRD patterns and shoudl be removed.
Reply: Thank you very much for the reviewer’s helpful comments. We have deleted in the article.

Reviewer 3 Report
General Comments
The work generates oxygen vacancies and Ti3+ defects on the surface of TiO2 nanoparticles with an argon plasma that reduces the band gap of TiO2 from 3.21eV to 3.17eV for Ar-TiO2 after treatment. The photocatalytic degradation of organic dyes by Ar-TiO2 is said to be enhanced relative to bare TiO2. Overall, this is an interesting manuscript but needs some more work to be completed for publication, including a few additional measurements. For that reason, the work is incomplete at this point. However, the manuscript should be encouraged for resubmission once the new measurements are produced and other improvements pointed below are addressed in the manuscript.
Major Comments
1) When introducing the theme of photocatalytic activity/efficiency in heterogeneous photocatalysis, the manuscript should also consider two related references that need to be discussed:
Hoque, Materials 2018, 11(10), 1990.
Murphy, Chem. Mater. 2015, 27, 4911
2) Connected to the previous point, the manuscript is missing work (i.e., to determine the photon flux) required to calculate the actual apparent quantum efficiency. Performing this missing measurement should allow the authors to provide a valuable piece of work.
3) The experimental section needs to provide the brand and purity of chemicals, etc. In addition, please provide some image/photo/diagram that helps to describe the setup employed. This will be especially helpful to the readers trying to visualize the measurements. The caption to figures with kinetics is also missing the very important “loading” of the photocatalyst employed in each experiment.
4) The following 3 key papers should be listed as recent contributing articles to hybrid photocatalysts in the references listed:
Xu, Applied Catalysis B: Environmental (2018), 230, 194-202.
Aguirre, Applied Catalysis B: Environmental (2017) 217, 485-493.
Chen, International Journal of Hydrogen Energy (2019), 44, 4123-4132.
Author Response
Reviews 3
1) When introducing the theme of photocatalytic activity/efficiency in heterogeneous photocatalysis, the manuscript should also consider two related references that need to be discussed:
Hoque, Materials 2018, 11(10), 1990.
Murphy, Chem. Mater. 2015, 27, 4911
Reply: Thank you very much for the reviewer’s helpful comments. We have carefully revised the manuscript and added the relevant literatures, i.e., ref. [6] and [7]. According to the relevant literature, photocatalytic hydrogen production experiments and photon flux detection were carried out. The apparent quantum efficiency was calculated by the obtained data. The detailed steps are shown in question 2.
In recent years, semiconductor photocatalytic technology has attracted much attention due to its excellent application prospects in the fields of sewage treatment, air purification, cleaning and sterilization, and solar energy conversion. [4-6] Semiconductor photocatalysis technology is an advanced technology that uses solar energy to carry out chemical reactions in a mild environment. [7-9]
[6]. Murphy, C. J.; Buriak, J. M., Best Practices for the Reporting of Colloidal Inorganic Nanomaterials. Chem. Mater. 2015, 27, 4911-4913.
[7]. Hoque, M.; Guzman, M., Photocatalytic Activity: Experimental Features to Report in Heterogeneous Photocatalysis. Materials 2018, 11 (10), 1990. DOi:10.3390/ma11101990.
2) Connected to the previous point, the manuscript is missing work (i.e., to determine the photon flux) required to calculate the actual apparent quantum efficiency. Performing this missing measurement should allow the authors to provide a valuable piece of work.
Reply: Thank you very much for the reviewer’s helpful comments.
The photocatalytic H2-production experiments were performed in a 300 mL sealed jacket beaker at ambient temperature and atmospheric pressure. In a typical photocatalytic experiment, 100 mg of Ar-TiO2 composite photocatalyst was suspended in 100 mL of aqueous solution containing methanol (20.0 V%) as sacrificial agents for trapping holes. Proper amount of H2PtCl6 aqueous solution was added in the above solution. Therefore, 3.0 wt% Pt, as a co-catalyst, was in-suit reduced during the photocatalytic hydrogen evolution reaction. Then, the reactor was bubbled with nitrogen for 30 min to completely remove the dissolved oxygen and ensured that the reactor was in an anaerobic condition. A continuous magnetic stirrer was applied at the bottom of the reactor in order to keep the photocatalyst particles in suspension status during the whole experiment. After 0.5 hours of irradiation, the chromatographic inlet was opened, and hydrogen was analyzed by gas chromatograph (GC9800, Shanghai Ke Chuang Chromatograph Instruments Co. Ltd, China, TCD, with nitrogen as a carrier gas and 5 A molecular sieve column). All glassware was carefully rinsed with deionized water prior to use. The apparent quantum efficiency (QE) was measured under the same photocatalytic reaction conditions. A Xe lamp source (300 W, 350-780 nm) was placed 10 cm directly above the reactor to serve as a light source to initiate a photocatalytic reaction. The focus intensity of the Xe lamp source and the beaker area are approximately. They are 800 mw·cm-2 and 30 cm2 respectively. Measure and calculate QE according to equation (1): [1, 2]
QE(%)=(number of reacted electrons )/(number of incident photons)×100
=(number of evolve H2 molecules×2)/(number of incident photons)×100 ……(1)
H2 precipitation experiments was carried out on Ar-TiO2 catalyst under full-spectrum conditions. After 2.5 h, the precipitation of H2 was 530 umol/g.h. number of incident photons: 4.615×1019. number of evolve H2 molecules:3.19×1019 Measure and calculate QE according to equation (1): QE=69%.
[1] Pan, D.; Han, Z.; Miao, Y.; Zhang, D.; Li, G., Thermally stable TiO2 quantum dots embedded in SiO2 foams: Characterization and photocatalytic H2 evolution activity. Applied Catalysis B: Environmental 2018, 229, 130-138.
[2] Hoque, M.; Guzman, M., Photocatalytic Activity: Experimental Features to Report in Heterogeneous Photocatalysis. Materials 2018, 11 (10), 1990. Doi:10.3390/ma11101990.
3) The experimental section needs to provide the brand and purity of chemicals, etc. In addition, please provide some image/photo/diagram that helps to describe the setup employed. This will be especially helpful to the readers trying to visualize the measurements. The caption to figures with kinetics is also missing the very important “loading” of the photocatalyst employed in each experiment.
Reply: Thank you very much for the reviewer’s helpful comments. In the previous work, we modified the catalyst with a low temperature plasma device to improve the performance of the catalyst. [1, 2]
(1) Experimental drug: (Ti (OC4H9)4), AR, Macklin; Ethanol (C2H5OH), AR, Fuyu.
(2) Experimental device:
Picture 1. Xe light source
Picture 2. Low temperature plasma device
(3) Experimental condition: Solution concentration: 10mg/L; organic solution: 100mL; catalyst: 50mg.
[1] Wang, Y.; Yu, F.; Zhu, M.; Ma, C.; Zhao, D.; Wang, C.; Zhou, A.; Dai, B.; Ji, J.; Guo, X., N-Doping of plasma exfoliated graphene oxide via dielectric barrier discharge plasma treatment for the oxygen reduction reaction. Journal of Materials Chemistry A 2018, 6, 2011-2017.
[2] Zhao, D.; Yu, F.; Zhou, A.; Ma, C.; Dai, B., High-efficiency removal of NOx using dielectric barrier discharge nonthermal plasma with water as an outer electrode. Plasma Science and Technology 2018, 20, 14020-014020
4) The following 3 key papers should be listed as recent contributing articles to hybrid photocatalysts in the references listed:
Xu, Applied Catalysis B: Environmental (2018), 230, 194-202.
Aguirre, Applied Catalysis B: Environmental (2017) 217, 485-493.
Chen, International Journal of Hydrogen Energy (2019), 44, 4123-4132.
Reply: Thank you very much for the reviewer’s helpful comments. We have added the relevant literatures, i.e., ref. [14], [15] and [30].
[14]. Xu, F.; Zhang, J.; Zhu, B.; Yu, J.; Xu, J., CuInS2 sensitized TiO2 hybrid nanofibers for improved photocatalytic CO2 reduction. Applied Catalysis B Environmental 2018, 230, 194-202.
[15]. Chen, W.; Wang, Y.; Shangguan, W., Metal (oxide) modified (M= Pd, Ag, Au and Cu) H2SrTa2O7 for photocatalytic CO2 reduction with H2O: The effect of cocatalysts on promoting activity toward CO and H2 evolution. International Journal of Hydrogen Energy 2019, 44 (8), 4123-4132.
[30]. Aguirre, M. E.; Zhou, R.; Eugene, A. J.; Guzman, M. I.; Grela, M. A., Cu2O/TiO2 heterostructures for CO2 reduction through a direct Z-scheme: Protecting Cu2O from photocorrosion. Applied Catalysis B: Environmental 2017, 217, 485-493.

Round 2
Reviewer 2 Report
The authors gave adequately addressed most of the reviewers’ comments. The manuscript has been improved and can be recommended for publication.
Author Response
The authors gave adequately addressed most of the reviewers’ comments. The manuscript has been improved and can be recommended for publication.
Reply: Thank you very much for the reviewer’s comments. We have carefully updated the manuscript.
Reviewer 3 Report
There is some improvement. However, none of the material from the response about the apparent quantum efficiencies is included in the manuscript. It appears the response has not experimentally determined the number of photons and the information came out of the blue. There is no information of the experimental procedure for this important determination in the manuscript. The apparent quantum efficiencies need to be reported.
Author Response
There is some improvement. However, none of the material from the response about the apparent quantum efficiencies is included in the manuscript. It appears the response has not experimentally determined the number of photons and the information came out of the blue. There is no information of the experimental procedure for this important determination in the manuscript. The apparent quantum efficiencies need to be reported.
Reply: Thank you very much for the reviewer’s helpful comments. We have made the following modifications to this problem. Also, we have carefully updated the language of the manuscript.
Page: 3 line: 128-154
2.4 Apparent quantum efficiency measurement:
The photocatalytic H2-production experiments were performed in a 300 mL sealed jacket beaker at ambient temperature and atmospheric pressure. In a typical photocatalytic experiment, 100 mg of Ar-TiO2 composite photocatalyst was suspended in 100 mL of aqueous solution containing methanol (20.0 V%) as sacrificial agents for trapping holes. Proper amount of H2PtCl6 aqueous solution was added in the above solution. Therefore, 3.0 wt% Pt, as a co-catalyst, was in-suit reduced during the photocatalytic hydrogen evolution reaction. Then, the reactor was bubbled with nitrogen for 30 min to completely remove the dissolved oxygen and ensured that the reactor was in an anaerobic condition. A continuous magnetic stirrer was applied at the bottom of the reactor in order to keep the photocatalyst particles in suspension status during the whole experiment. After 0.5 hours of irradiation, the chromatographic inlet was opened, and hydrogen was analyzed by gas chromatograph (GC7806, Beijing Shiwei Spectrum Analysis Instrument Co., Ltd. China, TCD, with nitrogen as a carrier gas and 5 A molecular sieve column). All glassware was carefully rinsed with deionized water prior to use. The apparent quantum efficiency (QE) was measured under the same photocatalytic reaction conditions. A Xe lamp source (300 W, 385 nm) was placed 10 cm directly above the reactor to serve as a light source to initiate a photocatalytic reaction. The focus intensity of the Xe lamp source is measured by a strong optical power meter (CEL-NP2000) and the focused area on beaker is 30 cm2. Measure and calculate the number of photons according to equation (1):
Measure and calculate apparent quantum efficiencies (AQE) according to equation (2): [50]
where h is the Planck constant (J·s ), c is Speed of light (m/s) and λ is the wavelength of linght (nm).
Page: 9 line: 293-300
The apparent quantum efficiencies (AQE) of the TiO2 and Ar-TiO2 catalysts was calculated by photocatalytic hydrogen production experiments under full-spectrum conditions. As shown in Table 4, the optical power density was 800 mW·cm-2 by the strong optical power meter measured. The number of incident photons were 4.615×1019 calculated by the Formula (1) calculated. As can be seen from Table 4, the sample of TiO2 exhibited a H2 evolution rates of 320 μmol·g-1·h-1 with a quantum efficiency of 41.7 % under UV light irradiation (λ = 385 nm) after 2.5 h. Compared with TiO2, the sample of Ar-TiO2 exhibited the highest H2 evolution rates (530 μmol·g-1·h-1) with a quantum efficiency of 69.0 %. It can be concluded Ar-TiO2 has higher light utilization efficiency.
Page: 11 line: 313-314
Table 4. Optical power density;the number of incident photons;the apparent quantum efficiencies (AQE) and H2 evolution rates( HER) of the TiO2 and Ar-TiO2 catalysts.
Page: 14 line: 446-447
50 Pan, D.; Han, Z.; Miao, Y.; Zhang, D.; Li, G., Thermally stable TiO2 quantum dots embedded in SiO2 foams: Characterization and photocatalytic H2 evolution activity. Appl. Catal. B: Environ. 2018, 229, 130-138.
